# Connexins and the Epithelial Tissue Barrier: A Focus on Connexin 26

**DOI:** 10.3390/biology10010059

**Published:** 2021-01-14

**Authors:** Laura Garcia-Vega, Erin M. O’Shaughnessy, Ahmad Albuloushi, Patricia E. Martin

**Affiliations:** Department of Biological and Biomedical Sciences, School of Health and Life Sciences, Glasgow Caledonian University, Glasgow G4 0BA, UK; v.lauramaria.garcia@gmail.com (L.G.-V.); Erinoshaughnessy@rocketmail.com (E.M.O.); albuloushi87@gmail.com (A.A.)

**Keywords:** epithelial tissue, connexin, gap junction, purinergic signaling

## Abstract

**Simple Summary:**

Tissues that face the external environment are known as ‘epithelial tissue’ and form barriers between different body compartments. This includes the outer layer of the skin, linings of the intestine and airways that project into the lumen connecting with the external environment, and the cornea of the eye. These tissues do not have a direct blood supply and are dependent on exchange of regulatory molecules between cells to ensure co-ordination of tissue events. Proteins known as connexins form channels linking cells directly and permit exchange of small regulatory signals. A range of environmental stimuli can dysregulate the level of connexin proteins and or protein function within the epithelia, leading to pathologies including non-healing wounds. Mutations in these proteins are linked with hearing loss, skin and eye disorders of differing severity. As such, connexins emerge as prime therapeutic targets with several agents currently in clinical trials. This review outlines the role of connexins in epithelial tissue and how their dysregulation contributes to pathological pathways.

**Abstract:**

Epithelial tissue responds rapidly to environmental triggers and is constantly renewed. This tissue is also highly accessible for therapeutic targeting. This review highlights the role of connexin mediated communication in avascular epithelial tissue. These proteins form communication conduits with the extracellular space (hemichannels) and between neighboring cells (gap junctions). Regulated exchange of small metabolites less than 1kDa aide the co-ordination of cellular activities and in spatial communication compartments segregating tissue networks. Dysregulation of connexin expression and function has profound impact on physiological processes in epithelial tissue including wound healing. Connexin 26, one of the smallest connexins, is expressed in diverse epithelial tissue and mutations in this protein are associated with hearing loss, skin and eye conditions of differing severity. The functional consequences of dysregulated connexin activity is discussed and the development of connexin targeted therapeutic strategies highlighted.

## 1. Introduction

Epithelial tissues line the outer surface of organs and the inner surface of cavities such as the digestive tract and secretory glands, where they project into the lumen connecting with the external environment. Thus, the epithelium separates tissue compartments forming barriers, regulates molecule exchange between those compartments and protects from biological, physical and chemical aggressions [1]. The integrity of the epithelium is maintained by intercellular junctional complexes composed of tight junctions (TJs), adherens junctions (AJs), and desmosomes [2,3]. These junctions aide in the formation of tight seals or barriers between the external environment. Since the epithelium is avascular, it is believed that the delivery and co-ordination of intercellular signals directly between cell layers is conducted via gap junction intercellular communication channels (GJIC) and paracrine signalling pathways [4,5].

Connexins (CXs) are the structural building blocks of gap junctions, four-transmembrane domain spanning proteins with intracellular N- and C-tails (Figure 1). In humans, connexins are encoded by a multigene family containing twenty-one members classified by their molecular weight ranging from 23 to 62 kDa in size (CX23-CX62) [6]. Six connexins oligomerise forming hemichannels or connexons linking the cytoplasm with the extracellular space. Two connexons from adjacent cells connect head-to-head to form an axial channel or gap junction, thereby allowing the interchange of ions and water-soluble molecules with a relative molecular mass up to 1.2 kDa. They play a central role in control of tissue development, homeostasis and a diverse range of cellular functions [7,8,9,10].

### The Epithelium and the Tissue Barrier

The structure of epithelial tissue depends on its function. In general, epithelial tissues are classified by their stratification, being simple, transitional or stratified. Simple epithelium composes surface-forming epithelia in contact with the basement membrane (e.g., epithelium of nephric tubules, trachea and secretory glands). The transitional epithelium has some cells in contact with the basement membrane and are surface-forming (e.g., epithelium of urinary bladder). Stratified epithelium is composed of a basal layer, in contact with the basement membrane and the only cells which can divide, and a superficial stratum, where the cells undergo two processes of differentiation: keratinisation non-cornification (e.g., epithelium of oral cavity, oesophagus, vagina) or keratinisation and cornification (e.g., epidermis, nail plate) [1]. The epithelium also plays a central role in body fluid secretions including heat and emotional responses via eccrine, apocrine and sebaceous glandular secretions that are tightly regulated and express a range of connexin proteins [11,12,13]. Skin appendages including the hair follicles and nails also extend from the stratified epidermis where connexins play a significant role in the hair follicle cycle and reviewed elsewhere [13,14]. This review will focus primarily on examples the role of CX26 and CX43 in epithelial tissue including stratified epithelium such as the epidermis and cornea and simple epithelium including the lining of the respiratory and intestinal tracts. Table 1 summarises the expression profile of CX26 in human tissue.

## 2. Connexins and the Skin

The skin is the human body’s largest organ and is composed of three layers: the hypodermis, dermis and epidermis. The hypodermis and the dermis are connective tissues, while the epidermis is epidermal tissue composed mainly of epithelial cells known as keratinocytes. Keratinocytes are subclassified into four layers: basal (proliferative cells), spinous, granular and corneous (anucleated squamous cells) [27]. The skin provides four different types of barrier: physical, redox, bio-chemical (innate immunity) and the adaptive immune barrier. The physical barrier consists of protein enriched cells and is mainly located in the stratum corneum and the granular layer, with strong adhesive interactions via tight, adheren and gap junctions. The bio-chemical or antimicrobial barrier consists of lipids, acids, lysozymes and antimicrobial peptides. These two barriers protect from external aggressions (outside-inside barrier), while also avoiding loss of water and solutes (inside-outside barrier) [28,29] (Figure 2).

The epidermis is composed by four layers of keratinocytes at different stages of differentiation and connexin pattern expression. The epidermis is a two sided barrier: the outside-inside barrier protects from environmental aggression, while inside-outside barrier protects from water loss.

The epidermis, like all other stratified epithelial tissues, is avascular so gap junctions play an important role in cell-to-cell communication and coordination. Up to 10 different connexins are differentially expressed throughout the epidermis, each presenting a characteristic expression profile dependent on the species and conditions. These Connexin profiles allow the establishment of specific gap junction communication compartments in the different strata. Alteration of connexin activity and compatibility may provoke problems in keratinocyte differentiation [5,30,31]. Several studies have demonstrated that connexins are involved in skin conditions such as chronic non-healing wounds, psoriasis and a variety of genetically related skin syndromes (e.g., [32,33,34]).

Within normal healthy human skin CX30.3, CX31 and CX43, and at lower levels CX26, C31.1, CX40 and CX45 are expressed in the granular layer. In the spinous layer, CX26, CX30, CX30.3, CX31, CX31.1, CX40, CX43 and CX45 are expressed with CX43 the predominant CX in the basal layer [5,30,35] (Figure 2). Cx26 is expressed at low levels in proliferating keratinocytes in tissue culture and readily expressed in stratified 3D epidermal cultures by immunocytochemistry [36,37,38].

### 2.1. Connexin 26, Trafficking and Assembly

CX26, a 26 kDa protein consisting of 226 amino acids, is encoded by the *GJβ2* gene, located on Chromosome 13q12.11 in Homo sapiens [39]. The gene is formed by the non-coding exon 1 (160 bp), an intron (3 kb) and exon 2 containing the complete connexin coding region and the subsequent 3′-UTR [40]. The promoter P1 (−128 bp:+2 bp) upstream of exon 1 has a number of transcription regulatory domains including SP1/SP3 and AP1 [39]. CX26 is also responsive to the NFkB transcription factor, which plays a role in inflammation and immunity, as well as in cell proliferation, differentiation and survival [41,42]. Human CX26 has a short C-terminal tail, with only 18 amino acids, this characteristic is relevant because it affects its interactome and post-translational modification, where CX26 interacts with a variety of proteins including those associated with the tight junction network [20]. Gap Junction assembly depends on the oligomerisation of Connexins to form a closed hexameric Connexon or hemichannel in the ER-Golgi environs, which are then escorted and inserted into the plasma membrane via the microtubule network, in association with motor proteins such as consortin [7,43,44]. Evidence suggests that CX26, and closely related CX30, can also follow an alternative Golgi-independent trafficking pathway, possibly providing a means for translation on free ribosomes and an ability to be rapidly translated at site specific plasma membrane locations when required [45,46,47,48,49,50]. Entire gap junction plaques are removed from the plasma membrane by the formation of annular gap junctions [4,51] (Figure 3). In addition, CX26 is unique to the connexin family as it does not contain phosphorylation sites in its C-terminal tail that play a significant role in protein turnover, particularly well characterised for Cx43 [51]. Nevertheless, CX26 presents a variety of putative post-translational modifications, including carbamylation [52], hydroxylation, phosphorylation and methylation, some of which happen at sites of deafness-causing mutations and may be associated with CX26 biogenesis and channel function [53,54]. Furthermore, hemichannels are normally closed during normal conditions; however, human CX26 hemichannels are an exception as they tend to be open under basal conditions. The 3D molecular structure of CX26 suggests this could be because, in contrast to other species, human CX26 presents an asparagine (uncharged) at amino acid position 159 in place of aspartic acid found in other species [55,56].

### 2.2. The Effects of CX26 Mutations

Many diseases have been linked with mutations in connexins, commonly termed connexin channelopathies [57,58,59,60]. CX43 is the most abundant connexin in the human skin; however, mutations and dysregulation of CX26, which is expressed at very low levels in healthy human epidermis, are related with skin disease characterised by abnormal keratinisation and hyperproliferation of the stratum corneum. Mutations in CX26 are among the most prevalent mutations associated with inherited non syndromic deafness (see Section 4.1) [61,62,63]. In addition to deafness, dominant mutations are also linked with a range of skin conditions of differing severity, suggesting a complex interrelationship between functional changes in connexin genotype and the phenotypic outcome [59,64,65,66] (Figure 1). Other mutations in the beta connexin subgroup including CX31 and CX30 cause similar epidermal dysplasia [59].

Mutations fall into four main classes. Class 1: trafficked to the plasma membrane with non-functional channels; Class 2: non-functional channels and protein trafficking deficiencies; Class 3: mutations associated with ‘leaky’ hemichannels and inflammatory skin disease; Class 4: trafficking deficiency and cell death, associated with mucositis, inflammatory disease and deafness. Class 1 mutations tend to be linked with non-syndromic deafness and have limited skin pathology and may be related with heterozygous advantage (see Section 4.1). Class 2 mutations align with non-inflammatory skin disorders such as Bart-Pumphrey and Vohwinkel syndrome and deafness. Vohwinkel syndrome (OMIM#124500) is a non-inflammatory disorder caused by CX26 mutations located predominantly on the first portion of EL1 (e.g., D66H). The disease is characterised by keratodermas with constriction bands around the phalanges, which induces autoamputation of the digits [67,68,69]. Class 3, gain of CX26 function mutations, are associated with inflammatory disorders such as Keratitis ichthyosis deafness (KID), Hystrix-like ichthyosis-deafness (HID). KID syndrome (OMIM#148210) is caused by CX26 mutations on the N-terminal tail, the EL1 (e.g., G12R, N14Y, G45E and D50N) [70,71,72]. In addition to hearing loss, the disease is characterised by: hyperkeratosis of the palms and soles, erythrokeratoderma on the extremities and face, follicular hyperkeratosis, photophobia and corneal vascularisation that ultimately leads to blindness. Patients experience severe and chronic bacterial and fungal skin infections and are susceptible to the development of squamous cell carcinoma [73,74]. The molecular mechanisms underlying the condition likely relate to “leaky” hemichannels, with differing sensitivities to pro-inflammatory mediators, and ionic sensitivity including calcium and zinc levels thereby altering channel function [66,75,76,77,78,79,80,81,82]. Several mutations induce lethal phenotypes and are associated with loss of cell viability [83,84,85]. Recently, we proposed a further Class 4 mutation group associated with hyperkeratosis, mucositis and deafness, where cell model studies revealed cell death and a collapse of the microtubule network, but limited connexin channel function (e.g., F142L, CX31G45E) [86,87].

Accumulating evidence suggests that, in addition to changes in CX26 trafficking and channel behavior, a key pathological trigger in the diverse CX26 mutations is alteration in CX oligomerization compatibility. Normally, oligomerization is CX subtype specific with ‘alpha’ and ‘beta’ subgroups being incompatible. As such, CX26 and CX43, critical connexins in epithelial tissue are unable to form heterotypic structures [88,89]. Recent studies report changes in CX26 mutation oligomerization compatibility, allowing aberrant interactions with CX43 with exacerbated hemichannel activity but non-functional gap junction channels [78,86,90], with each mutation uniquely altering the 3D structure in terms of charge, pore size, hydrophobicity, etc. It is also conceivable that altered CX43:CX26 heteromeric channels influence unique metabolic exchange and influence asymmetric cell division required for the stratification of the epidermis, thereby contributing to the hyperproliferative status of the skin [91]. Further characterisation and understanding of the impact of such aberrant CX signalling is required to enhance understanding of these complex conditions. Interestingly, a recent report by Laird and colleagues suggest that hearing loss caused by CX26 mutations does not depend on any interaction with CX43 [92].

### 2.3. The Effects of CX26 Dysregulation

Up-regulation of CX26 is a characteristic of hyperproliferative epidermis in physiological conditions, such as vaginal and buccal epithelium, which shows a high proliferation rate, and pathological conditions, such as psoriatic epidermis, chronic non-healing wounds epidermis and viral warts [93,94,95,96]. CX26 expression is also induced during wound healing and skin hyperplasia stimulated by tumor promoters [40]. Transgenic mice over-expressing CX26 in the suprabasal layer developed a hyperproliferative phenotype, providing models for several epidermal human CX26 diseases, such as psoriasis [97].

#### 2.3.1. Connexins and Wound Repair

Connexins are closely involved in normal wound repair and show dynamic changes in expression after wounding. After 6 h of injury, CX43 is down-regulated in keratinocytes and fibroblast at the wound edge to allow cell migration followed by recovery of CX43 levels [95]. In contrast, CX26 and CX30 are up regulated in keratinocytes until the wound is closed [96,98,99] in the granular cell layer near the wound margins and in the basal cell layer at some distance from the wound [95]. Thus, Connexins play a pivotal role in a variety of aspects of the acute wound healing process and each step is associated with a different connexin environment [51].

At the edge of chronic wounds, epidermal CX43, CX26 and CX30, and dermal CX43 are strikingly up-regulated around wound margins and at some distance from the chronic wound [95]. The up-regulation of CX43 may disrupt fibroblast migration to the wound bed and as a result failure of granulation tissue formation occurs [100,101,102,103,104]. The up-regulation of CX26 may contribute to the inflammatory and hyperproliferative status of the wound [95,105]. The overexpression of Cx26 in mouse skin (under the control of the involucrin promoter) kept wounded epidermis in a hyperproliferative state, blocked the transition to remodelling, and led to an infiltration of immune cells. This overexpression also induced ATP release from keratinocytes, which delayed epidermal barrier recovery and promoted an inflammatory response in resident immune cells [97].

#### 2.3.2. Connexins and Psoriasis

Psoriasis is a chronic hyperproliferating skin disorder that manifests sporadic skin lesions characterised by loss of the granular layer and incomplete keratinocyte differentiation associated with a thickened cornified layer. Overexpression of CX26 in psoriatic lesions was originally reported by Labarthe and Lucke in the late 1990s [93,94]. Subsequent studies in mouse models overexpressing Cx26 in the skin showed a mildly acanthotic and hyperkeratotic skin, with a thicker and more compact cornified layer. Areas with frictional trauma, such as axillary areas, presented with scaling and desquamation and hyperkeratotic plaques developed [97]. Other studies revealed that tape stripping of normal human epidermis induced CX26 expression and hyperproliferation [106]. More recently, publication of the psoriatic transcriptome permitted in depth RNAseq analysis [107] and many of the upregulated genes are related with cell-to-cell adhesion complexes. *GJB2*, encoding CX26 was the 98th most up-regulated gene detected and its over expression is used as a marker of genetic predisposition in psoriasis [108,109]. A variety of other transcriptomic studies confirm this overall increase in CX26 expression in psoriatic tissue yet no changes in CX43 gene expression are reported [110,111].

## 3. Connexin 26 and Other Epithelial Tissue: A Link with Heterozygous Advantage

### 3.1. Connexin 26 in Intestinal Epithelia

The intestinal epithelium is lined with a single layer of polarized cells where the tight junction network, located at the interface between the apical and basolateral membranes, plays a pivotal role in maintaining the mucosal barrier and regulating selective paracellular transport [112]. The intestinal connexins display cell type-specific expression patterns, delineating physiological compartments with specific functions. Up to 11 different isoforms are expressed in the intestinal system: CX26, CX32, CX37 and CX43 are notably expressed in the epithelial cells of the small intestine and colon [17]. CX26-related GJIC plays a role in the maintaining epithelial barrier function by affecting the production of TJ proteins. For example, in Caco-2 cells, a human colorectal epithelial cell line, CX26 expression increased gradually while cells reached confluency. Overexpression of CX26 resulted in enhanced tight junction formation with lower levels of mannitol flux and increased claudin-4 protein expression [113]. Recent studies also indicate that the CX26 interactome includes components of the tight junction network [20].

### 3.2. Connexin 26 in the Auditory System

As mentioned in Section 2.2, mutations in CX26 are a leading cause of deafness. The auditory canal surface is covered by a simple nonkeratinized squamous epithelium continuous with the lining of the tympanic cavity where CX29 is the predominant connexin expressed. This protein does not form functional gap junctions and is believed to act as a hemichannel in association with voltage-dependent K^+^ channels [114]. CX26 is expressed in the organ of Corti, the spiral limbus, the stria vascularis, and fibrocytes of the spiral ligament [114]. The intrical auditory connexin circuit is critical in maintaining the recycling of K^+^ ions in the endolymph enabling auditory function. As discussed multiple recessive mutations in CX26 are associated with hearing loss with a relatively high carrier frequency (1–3%), suggesting the possibility of heterozygous advantage [115,116,117,118,119,120,121,122]. Within different ethnic groups there are specific common recessive mutations that account for most of the CX26-related hearing loss, e.g., 35delG, 235delC and R143W, a typical ‘class 1 mutation’ in the European, Japanese and African populations, respectively [116,117,118,122].

### 3.3. Connexin 26 and Heterozygous Advantage

The mutations CX26-R134W and 35delG are reported to produce a thicker epidermis [119,120]. Organotypic co-culture skin models with keratinocytes expressing CX26-R134W showed a thicker epidermis, suggesting an increase in cell proliferation, a delay in terminal differentiation, and higher cell migration than CX26WT-models, possibly reflecting a greater protection against environmental damage [120]. CX26 is also expressed in intestinal epithelial cells and several studies suggest that ‘Class 1’ heterozygous CX26-mutation status may decrease susceptibility to enteric pathogens. For example *Shigella flexneri* causes bacillary dysentery in humans and it invades intestinal epithelial cells inducing an acute inflammatory reaction destroying the colonic epithelium [123]. This pathogen can open CX26 hemichannels, in an actin-phospholipase C-dependent manner, allowing ATP release, which induces the retraction of filopodial extensions, subsequently bringing the bacteria into contact with the cell body being invaded [124]. In vitro experiments with human intestinal cell lines showed a reduction of both cellular invasion by *S. flexneri* and adherence by enteropathogenic *E. coli* following treatment with CX26 siRNA, suggesting that the loss of functional CX26 provides protection against gastrointestinal bacterial pathogens [125]. Further population studies on the health of CX26 carriers in relation to epithelial disorders are certainly warranted, with recent evidence suggesting a link between CX26 and psoriatic hearing loss [126].

### 3.4. The Airway Epithelium

The air-conducting portion of the lung is lined by a pseudostratified epithelium, which is referred to as the airway epithelium. CX26 and CX43 are expressed in the undifferentiated human airway epithelium in conductive airways, but upon differentiation they rapidly disappear. As with respiratory airways, CX26 and CX43 have been detected in human lung epithelial cells [127,128]. Upon wounding, CX26 expression is transiently induced in activated basal cells and strongly decreased after wound closure. Activation of cell proliferation upon injury is required to trigger CX26 expression in repairing cells. Thus, induction of CX26-mediated intercellular communication by proliferative signals in repairing basal cells may represent a feedback mechanism to repress their proliferation and progressively promote differentiation [129]. Other studies suggest a role for CX26 in barrier adhesive complexes. For example, Calu-3 cells, a human transformed bronchial epithelial cell line, do not express CX26. Treatments with ouabain, a Na^+^/K^+^-ATPase inhibitor, disrupted barrier function, causing a down-regulation of occludin, JAM-1, claudin-2, and -4 expression and up-regulation of ZO-1 and claudin-14. However, when the cells were transfected to express CX26 they were not affected by ouabain, TJ proteins remain unchanged and CX26 co-localised with claudin-14. Pre-treatments with GJIC inhibitors did not affect CX26 changes, suggesting that CX26 expression but not GJIC regulates the TJ barrier in human epithelial cells [130]. Some of these concepts were recently reviewed by Chanson et al., 2018 [5].

### 3.5. The Cornea

The cornea is a transparent avascular tissue that acts as a barrier protecting the eye from infections. The corneal epithelium is composed of 5–7 layers of three types of non-keratinised stratified squamous cells: superficial cells, wing cells and basal cells [131]. CX26, CX30, CX31.1 and CX43 are the predominant connexins in human cornea [132,133], their mutation results in ocular disorders such as oculodentodigital dysplasia, produced by mutations in CX43 and KID syndrome related to CX26 mutations, where keratitis is a major pathology [73,134,135]. Zhai et al. determined the connexin expression in 10 human diseased corneas: five injured by chemical burns and five infected [133]. They determined by flow cytometry that only 0.5% cells in normal corneal tissue expressed CX26; however, this percentage increased in chemical burn and infected corneas, being 15.6 and 34.2%, respectively. On the other hand, CX43 was expressed in 3.1% of normal cells, while its expression was 23.4 and 40% in chemically burned and infected cells [133]. As such, Connexins emerge as a prime therapeutic targets and treatments with the synthetic CX43 mimetic peptides and SiRNA technologies are proving successful in clinical trials (see Section 5) [136,137,138].

## 4. Connexins and Inflammation

### 4.1. Molecular Mechanisms: ATP, Ca^2+^ and the Pro-Inflammatory Response

Formation and maintenance of barrier function is affected by cytokines, ATP and Ca^2+^ gradients throughout the stratified layers of the epidermis. Ca^2+^ levels are highest in the granular layer and almost disappear in the stratum corneum and is an important regulator in epidermal differentiation, protein synthesis and cell-to-cell adhesion [28,139]. Furthermore, keratinocytes in each layer of the epidermis show distinct characteristics in response to ATP. Purinergic signalling, an extracellular signalling pathway triggered by purine nucleotides including ATP passively released via hemichannels, participates in multiple physiological processes (e.g., proliferation and differentiation), activates nerve cells and attracts immune cells. Dysregulation of ATP results in pathological conditions including chronic inflammation. Keratinocytes release ATP in a critical gradient depending on the differentiation status of the cell and the distribution of purinergic receptors is different in each layer of the epidermis, thus the response to ATP is different in different epidermal compartments [140,141]. Exposing keratinocytes to stress induced both elevation of intracellular Ca^2+^ and ATP release [141]. Furthermore, the use of gap junction blockers significantly reduced Ca^2+^ wave propagation in differentiated keratinocytes, suggesting the participation of gap junctions in the induction of Ca^2+^ waves in response to stress in the epidermis [140,142]. A wide variety of stimuli, including mechanical stimulation, induce ATP release through hemichannels that subsequently interacts with purinergic receptors promoting the propagation of Ca^2+^ signals, from the ER. These signals can be directly propagated to neighbouring cells via gap junction channels [143] or through the extracellular space via hemichannels [34,144]. The increase in intracellular Ca^2+^ in response to ATP varies in each layer of the epidermis and is reported to be higher in the basal than in the outer layers [141]. Such events enable a rapid and co-ordinated spatio-temporal signal propagation throughout tissues.

Human CX26 hemichannels can be opened under resting conditions [55]. Mutations in CX26 affect hemichannel permeability. As discussed, CX26 KID syndrome mutations provoke ‘leaky’ hemichannels and allow the aberrant formation of CX43-CX26 hetero-hemichannels presenting abnormal permeability [66,78,86,145,146,147]. Some of these mutations also alter CX26 PCO_2_ gating sensitivity, which may contribute to CO_2_-dependent regulation of breathing in mammals [148,149,150,151]. Other examples of dysregulation of epidermal calcium gradients were observed in a transgenic KID syndrome mouse model, Cx26-S17F, where intra- and extracellular Ca^2+^ levels were maintained in the corneous layer, proposed to be a consequence of hyperactivation of heteromeric hemichannels. The epidermis of these mice had altered lipid profiles, developed both hyperproliferation and hyperkeratosis which negatively affected epidermal water barrier [152].

TNFα, IL-1 and IL-6 are potent mitogens and stimulators of lipid synthesis crucial to respond to and repair barrier disruption; however, the chronic expression of these cytokines lead to inflammation and epidermal proliferation, an end result in epidermal dysplasia including conditions such as dermatitis and psoriasis [28]. ATP induces IL-6 production in HaCaT cells, a model keratinocyte cell line, via P2Y receptors, and it is critical for the wound repair process. Mechanical stimulation of HaCaT cells, such as the change of medium or silica nanoparticle 30, induced release of ATP and the activation of P2Y receptor provoked an increase in intracellular Ca^2+^ concentration causing the induction of IL-6 mRNA and protein levels [144,153]. This response was inhibited by suramin (a P2 purinergic receptor antagonist) [154,155].

The deregulation and persistent production of IL-6 contributes to chronic inflammation and autoimmunity. An example is psoriasis, of which transcriptome analysis determined that IL-6 is up-regulated 4.3-fold in psoriatic lesions [107]. In addition, ATP increases proliferation in HaCaT cells and maybe a mechanism for the hyperproliferation that occurs in psoriasis. Both, IL-6 production and proliferation in HaCaT cells were inhibited by blocking purinergic receptors with blockers such as suramin [154,155,156].

There is extensive evidence to suggest that a double positive feedback between connexin channels and purinergic signalling exists. The opening of connexin channels allows the release of ATP which is an agonist of purinergic signalling leading to the opening of hemichannels and the subsequent amplification of purinergic signalling (more ATP release and Ca^2+^ waves) via a positive signalling feedback loop, giving rise to the concept of ATP-induced ATP release [157]. Mugisho and colleagues propose the inflammasome pathway is amplified in an autocrine manner through ATP released via CX43 [158]. Evidence builds that other connexins, including overexpression of CX26 in the epidermis may contribute to this cycle of events with ATP released feeding into purinergic signalling pathways and associated with exacerbated inflammation and hyperproliferation linked with psoriasis.

### 4.2. Connexins and Gram-Positive Bacteria

Study of the microbiome have suggested a further contributing trigger to Psoriasis is as a dysbiosis from commensal microflora colonisation (e.g., *Staphylococcus epidermidis*) to opportunistic colonisation (e.g., *S. aureus*), i.e., an ‘outside-in’ trigger [159,160]. Such a microbiome shift would lead to altered innate immune signalling patterns and may contribute to the pathogenesis. A similar shift is also observed in KID syndrome and chronic wounds where connexin dysregulation is also evident [73].

Peptidoglycan (PGN) from *S. aureus* regulates connexin gene and protein expression in diverse tissues. In the HaCaT keratinocyte cell model system it induced CX26 gene expression [42,81]. A similar response was observed in mouse astrocytes; however, in microglia and endothelial cells increased Cx43 gene and protein levels were observed [161,162,163]. PGN is a potent inducer of the innate immune response recognised by Toll-like receptor 2 (TLR2), which is the prime receptor for PGN [164]. PGN interaction with TLR2 triggers a downstream signalling cascade by initiating Ca^+2^ fluxes which finally activates NF-kβ and, subsequently, pro-inflammatory cytokine expression such IL-6, IL-8, TNF-α and IL-1β, [165,166,167]. Gap junctions are involved in the initial spread of Ca^+2^ fluxes generated by TLR2 in mucosal epithelium. Further, connexin-mediated pathways transduced the signalling triggered by an infection and provoked gene expression changes including cytokine expression. For example, *Pseudomonas aeruginosa* activated TLR2 and triggered a Ca^2+^ wave, that passed through gap junction channels, subsequently activating NF-kβ and inducing IL-8 secretion in airway cells [165,166,167]. Furthermore, in HaCaT cells exposed to PGN from *S. aureus*, there was an increase in ATP release and TLR2 expression. TLR2, via NF-kB, induced the up-regulation of CX26, IL-6 and IL-8, and this response was inhibited by connexin channel blockers [42]. In other studies, PGN evoked hemichannel activity and induced a pro-inflammatory response in HaCaT cells expressing class 3 CX26 mutations associated with KID syndrome (Section 2.2), events that were inhibited by carbenoxolone, a connexin channel inhibitor. This did not occur in cells expressing non- functional mutations such as D66H associated with the non-inflammatory skin disorder Vohwinkel syndrome [81].

## 5. Future Directions and Connexins as Therapeutic Targets

Dysregulation of connexin expression is thus associated with a wide range of epithelial tissue events. This tissue, particularly the stratified epidermis and the cornea, is readily accessible and prime for therapeutic intervention [168,169]. A range of connexin mimetic peptide and SiRNA technologies continue to be developed targeting connexins with evidence amounting of their considerable therapeutic benefit, most of which are based on CX43 [170,171]. Initial studies included antisense oligonucleotides targeted to CX43 [172,173,174]. This was successfully taken forward by Professor Colin Green and colleagues to clinical trials with exciting evidence in wound healing scenarios (https://ocunexus.com/nexagon). Recently antisense technologies targeting CX26 were reported to reduce pro-inflammatory ATP release within the skin and were beneficial in in vivo wound healing studies, preventing CX26 upregulation at the wound edge, reducing inflammation, epidermal thickening and improving wound closure events [96]. Short connexin mimetic peptides targeting CX43, based on the original work pioneered by Professor Howard Evans and colleagues during the 1990s, provided a platform for peptide design effectively inhibiting both Gap Junction and Connexin hemichannels activity. The ‘GAP27’ domain targets amino acid sequence SRPTEKTIFFI located on the distal domain of EXL2. This sequence is highly conserved between Cx43 and Cx37 with a variety of studies revealing Connexin subtype specificity [37,175,176,177,178]. Various platforms have now taken this work forward with the success of extracellular loop peptides, analogues of GAP27, targeting CX43 being successful in clinical trials for patients with ocular conditions including diabetic retinopathy and age related macular degeneration [179,180] (https://ocunexus.com/peptagon). Although there is currently no published information on Cx26 specific peptides, it is likely that targeting this domain could yield peptides capable of uniquely blocking Cx26 activity. The peptide ACT-1, targeting the carboxyl tail region of CX43 shows exciting possibility in phase 3 clinical trials in wound healing and in tailored targeting of cardiac disease by Prof Rob Gourdie and colleagues [181,182] (https://firststringresearch.com/).

Finally, allele specific SiRNA therapies, targeting specific CX26 mutations, show early promise in advancing personalised medicine approached for patients with conditions such as KID syndrome [183,184].

In conclusion, Connexins are dynamically expressed within epithelial tissue with CX26 being highly responsive to environmental stimuli. Further studies are warranted on the interplay of the CX26:CX43 nexus in epithelial networks and may enable tailored targeting of both Connexins to resolve a range of pro-inflammatory epithelial conditions.

## Figures and Tables

**Figure 1 biology-10-00059-f001:**
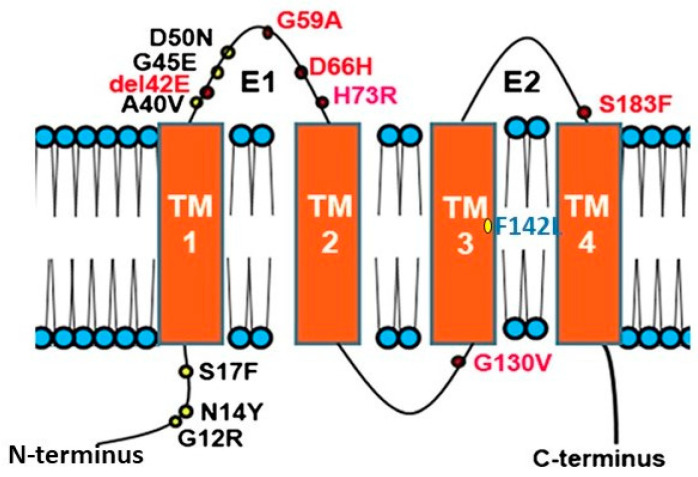
Topology of Connexin 26 and examples of mutations in CX26 associated with epidermal dysplasia and hearing loss. N amino terminus; E1 extracellular loop 1; E2 extracellular loop 2; TM transmembrane domain. Mutations in red indicate ‘Class 2′ black, ‘Class 3′ and blue ‘Class 4′ mutations (see Section 2.2).

**Figure 2 biology-10-00059-f002:**
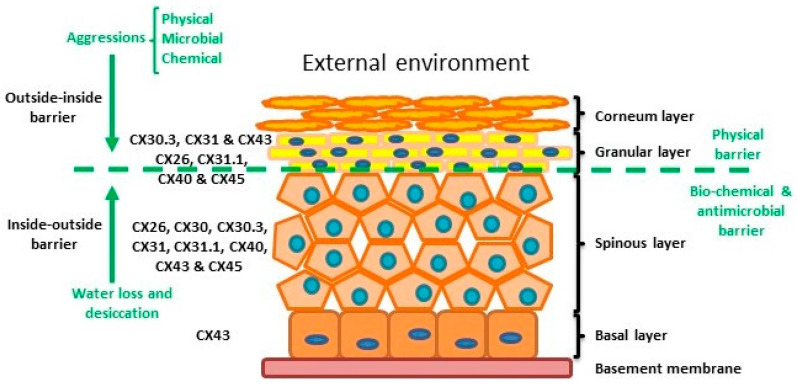
Structure of the epidermis indicating connexin expression profile and barriers.

**Figure 3 biology-10-00059-f003:**
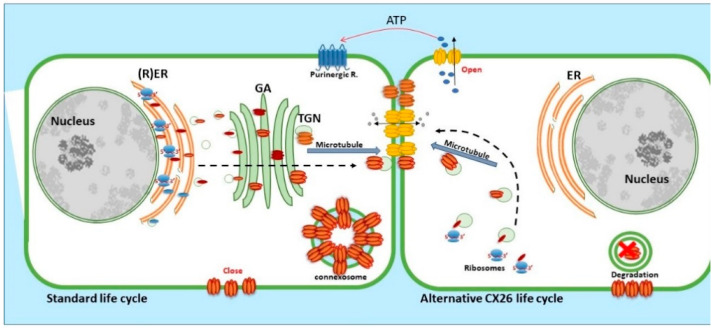
Life cycle of CX26: CX26 can be co-translated and trafficked through the secretory route (standard life cycle). It can also be post-translationally incorporated into ER microsomes and trafficked by Golgi-independent pathway (alternative CX26 life cycle). GJIC mediated communication is indicated between cells and paracrine signalling permitting ATP release indicated via the open hemichannels.

**Table 1 biology-10-00059-t001:** Expression of CX26 in human tissue.

System or Organ	Tissue or Structure	Cell Type	References
Skin	Epidermis	Keratinocyte	[13,15]
	Spinous layer		
	Granular layer		
Appendages	Sebaceous gland		[13,15]
	Eccrine sweat gland and ducts		
	Hair follicle.		
	Outer root sheet		
	Inner root sheet: Henley and Huxlye		
	Hair shaft: Cortex and Medulla		
	Matrix		
Brain	Occipital cortex	Astrocyte (glia)	[16]
	Diencephalon	Leptomeningeal cells	
	Medulla oblongata		
	Caudate nucleus		
Digestive system	Stomach	Epithelial cells	[17]
	Small intestine	Muscularis externa cells	
	Colon		
Endocrine and exocrine glands	Salivary glands	Acinar cells	[18,19,20]
	Pancreas (serous acini)	Beta cells	
	Pituitary (adenohypophysis)		
	Parathyroid	Principal cells	
	Thyroid (follicles)		
	Preputial ducts		
	Lacrimal (serous acini)		
	Parotid (serous acini)		
	Liver	Periportal hepatocytes	
Kidney	Proximal tubule		[21]
Reproductive system	Endometrium (luminal epith.)	Basal glandular cells	[22,23]
	Preimplantation embryo	Blastocysts	
	Placenta	Syncytiotrophoblasts	
	Myometrium	Uterine myocytes	
Endocrine system	Adrenal cortex		[24]
Ear	Spiral limbus	Fibrocytes	[25]
	Spiral ligament		
	Striavascularis		
	Cochlea	Claudius cells	
		Hensen’s cells	
		Inner sulcus cells	
Lung	Alveolar epithelium		[26]

## Data Availability

Not applicable.

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
