# Peer review of "Connexins and the Epithelial Tissue Barrier: A Focus on Connexin 26"

_biology, 2021, doi:10.3390/biology10010059_

Round 1
Reviewer 1 Report
This manuscript is a review summarizing the literature on roles for connexins in epithelial function. By and large the review covers the topic well, is timely and will be of general benefit to the field. I have only a few concerns/suggestions
Considering the theme of the manuscript, the authors should consider adding a separate section, including additional primary literature, that specifically covers roles for connexins in epithelial barrier function (e.g. tight junctions). There is some coverage of this in a few places, but this could have been more thoroughly presented.
Line 13: Some epithelia are not topologically exposed to the outside environment. Consider changing this sentence
Lines 31-32: "Connexin 26, the smallest member of the connexin family" Cx26 is 226 amino acids long, Cx23 (as mentioned on line 55) is 203 aminio acids long. Consider revising the text to be more precise. This may also be an issue on lines 109-111 and in other places in the manuscript.
Lines 50-51: Paracrine signalling is another possible mechanism mediated by connexin hemichannels. This could be expanded throughout the manuscript.
Lines 69-79: The section of epithelial tissue diversity is very brief. Consider expanding this section and including a diagram or removing it.
Section 4.1. It's unclear why the intestine and auditory system are combined like this. Consider having a separate (and more expanded) section describing roles of Cx26 in hearing, which has been very well studied. If the gut epithelium is not well suited to stand alone, it could be combined with airway epithelium (section 4.2).
Sections 5 and 6 could be combined into a more general section on connexins and inflammation with subheadings.
Consider expanding section 7 on therapeutics to mention putative mechanisms of action for the different drug candidates. More specific examples and citations would significantly strengthen the conclusions of the manuscript. Also, the focus of the manuscript is predominantly Cx26, but all the drug candidates mentioned target Cx43. Are there any candidates that target Cx26?
Author Response
We would like to thank the reviewer for their helpful comments and have summarised our response to comments below in italics for clarity.
This manuscript is a review summarizing the literature on roles for connexins in epithelial function. By and large the review covers the topic well, is timely and will be of general benefit to the field. I have only a few concerns/suggestions
Considering the theme of the manuscript, the authors should consider adding a separate section, including additional primary literature, that specifically covers roles for connexins in epithelial barrier function (e.g. tight junctions). There is some coverage of this in a few places, but this could have been more thoroughly presented.
Line 13: Some epithelia are not topologically exposed to the outside environment. Consider changing this sentence
This sentence has been revised
Lines 31-32: "Connexin 26, the smallest member of the connexin family" Cx26 is 226 amino acids long, Cx23 (as mentioned on line 55) is 203 aminio acids long. Consider revising the text to be more precise. This may also be an issue on lines 109-111 and in other places in the manuscript.
This has been revised, we have not commented extensively on Cx23 and Cx25 but recognise that these are indeed smaller connexins. As not very much is known about these we have revised to state that Cx26 is one of the smallest. The overall size is now stated.
Lines 50-51: Paracrine signalling is another possible mechanism mediated by connexin hemichannels. This could be expanded throughout the manuscript.
We have included paracrine signalling pathways to reflect hemichannel activity and this is further stated throughout the manuscript.
Lines 69-79: The section of epithelial tissue diversity is very brief. Consider expanding this section and including a diagram or removing it.
We have left this short paragraph but included a statement that links with the following sections
Section 4.1. It's unclear why the intestine and auditory system are combined like this. Consider having a separate (and more expanded) section describing roles of Cx26 in hearing, which has been very well studied. If the gut epithelium is not well suited to stand alone, it could be combined with airway epithelium (section 4.2).
The rationale for merging this section was to draw attention to the evidence linked with heterozygous advantage. While we appreciate the suggestion to extend the section on auditory function we felt that this is summarised in a variety of recent reviews. We have added a few extra sentences which we trust creates a better link.
Reviewer 2 Report
In this manuscript, Garcia-Vega et al. provide a comprehensive review of the role of connexins in the epithelial barrier. In general, the authors addressed a contemporary topic of interest to general readers. It covers most of the recent findings in the related field. The manuscript, in most parts, is well written and organized. As for the weakness of the manuscript, the factors and molecular mechanisms involved in the regulation of connexins, especially CX26 in epithelial tissues, have not been clearly described. Besides, recent development about the effects and molecular mechanisms of connexins in the control of several key molecular events in inflammatory responses, such as NFkb activation and inflammasome formation, should also be mentioned. Lastly, typos and spacing errors (such as in lines 162 and 166) should be corrected. Some sentences are difficult to understand (such as lines 168-169). Additional polishment on writing is recommended.
Author Response
Reviewer 2
We would like to thank the reviewer for their helpful comments and have summarised our response to comments below in italics for clarity.
In this manuscript, Garcia-Vega et al. provide a comprehensive review of the role of connexins in the epithelial barrier. In general, the authors addressed a contemporary topic of interest to general readers. It covers most of the recent findings in the related field. The manuscript, in most parts, is well written and organized. As for the weakness of the manuscript, the factors and molecular mechanisms involved in the regulation of connexins, especially CX26 in epithelial tissues, have not been clearly described. Besides, recent development about the effects and molecular mechanisms of connexins in the control of several key molecular events in inflammatory responses, such as NFkb activation and inflammasome formation, should also be mentioned. Lastly, typos and spacing errors (such as in lines 162 and 166) should be corrected. Some sentences are difficult to understand (such as lines 168-169). Additional polishment on writing is recommended.
Response reviewer 2
We would like to thank the reviewer for their supportive comments. The typos have been corrected and the manuscript further proof read.
With regard to Nf-kB pathway and the inflammasome are referred to in section 5.1 and 5.2
Reviewer 3 Report
This is a comprehensive, well written review about the physiological and pathophysiological role of connexins in epidermal tissues. The main focus of the review is on diseases associated with mutations or dysregulation of connexins particularly Cx26. The review also discusses recent advances in connexin targeted clinical therapeutic strategies. The review does a good job of covering most of what is known in this area.
Specific comments:
I found a number of minor typos. For example, in the Simple Summary:
line 17. epithelial
lines 21-23. awkward sentence
Abstract
line 24. Epithelial tissues
line 32. epithelial tissue
Figure 1, p.2. Why are some of the mutations in written in red?
p. 4, line 128. Figure 3 should be referred to earlier in the paragraph. Its not related to the 3D molecular structure of Cx26.
p. 6, line 199. mouse skin
Author Response
We would like to thank the reviewer for their helpful comments and have summarised our response to comments below in italics for clarity.
This is a comprehensive, well written review about the physiological and pathophysiological role of connexins in epidermal tissues. The main focus of the review is on diseases associated with mutations or dysregulation of connexins particularly Cx26. The review also discusses recent advances in connexin targeted clinical therapeutic strategies. The review does a good job of covering most of what is known in this area.
Specific comments:
I found a number of minor typos. For example, in the Simple Summary:
line 17. epithelial
lines 21-23. awkward sentence
Abstract
line 24. Epithelial tissues
line 32. epithelial tissue
Figure 1, p.2. Why are some of the mutations in written in red?
- 4, line 128. Figure 3 should be referred to earlier in the paragraph. Its not related to the 3D molecular structure of Cx26.
- 6, line 199. mouse skin
We would like to thank the reviewer for their supportive comments. The typographical errors have been modified and clarification on points above revised.
Figure 1 has been revised to highlight the different mutations
Reviewer 4 Report
Garcia-Vega et al. review the role of connexins and the epithelial barrier with a focus on Cx26. It is not made clear why the authors have chosen to focus on Cx26, especially since the review is overall very general and only very briefly touches upon several broad topics that concern many connexin isoforms. Consequently, the review serves mostly as a very general introduction to the topic as it does not provide too much deep insight into a specific topic or problem or provide a strong point of view or gives a unique insight. Indeed, the general nature of the article is perhaps underscored by the fact that 19 of the first 20 references are reviews themselves.
Some minor suggestions and some moderate issues should be addressed to improve the purpose as a general review that serves to introduce someone to the topic.
In general, the authors focus on Cx26 and “barrier” function. Barrier function needs to be explained in more detail (including the molecular mechanism regulating this phenomenon) as well as the direct evidence that Cx26 truly contributes to barrier function, or not.
Specifics:
Line 13: rephrase:
The wording “external environment” may be misleading to the reader (it may not be obvious this only refers to external to the organ itself). Consider rephrasing keeping in mind that epithelial cells cover the inner surface of the internal organs and the outer surface of the human body.
Line 17: For clarity consider deleting or moving “Within the epithelia” (no “L” at the end) to after “protein function”
Line 31: rewrite. Purinergic signaling is not a “physiological process”. (Simplest rewrite would be “…wound healing VIA purinergic signaling…”)
Line 32: Correct the erroneous statement that Cx26 is the smallest member of the family.
Line 33: Add that Cx26 is also associated with keratitis (eye disease of another type of epithelia).
Line 44: rephrase confusing sentence considering epithelial (endothelial) cells are lining what is considered the “inner” surface of blood vessels.
Line 49: rephrase to avoid confusion. Although gap junctions would play an important role in avascular epithelia, other pathways are providing for example oxygen and glucose delivery, it is not a gap junction-exclusive exchange of signals.
Line 62-78: Section 2 is a short and very general paragraph that perhaps could be included in the general introduction that describes the epithelium. Or perhaps expand the section. In any case, highlight that there are many additional important functions of the epithelium apart from barrier function (e.g. glandular secretion).
Line 83: Specify to the reader that keratinocytes are epithelial.
Line 109 onwards: In the section of connexins in the skin, the expression of Cx26 is not very well detailed. Is Cx26 truly expressed in normal non-irritated skin? or only palmoplantar skin? What about Cx26 in the keratinocytes in hair follicles and sebaceous gland epithelial cells?
The section that follows on translational regulation and trafficking (and associated Figure 3) is not particularly relevant to the topic or the section “connexins in the skin”. It would valuable to add some information on the actual transcriptional regulation (eg AP1 and other transcription factors) that underlies the dramatic upregulation of Cx26 under certain situations. This can be linked to certain inducing pathways such as interleukins etc. as discussed later.
Line 110: Confirm whether Cx26 has the shortest C-tail (shorter than Cx23 and Cx25?)
Line 124-6: the phrases “standard” and “resting” conditions are not ideal.
Line 145 onwards. Are the four classes of mutations a classification suggested by the authors? Specify and if not provide a reference. What is determining these mutation classes? The molecular effect (eg trafficking) or the phenotype? Or both? Which class does for example R143W, which causes reduced cell death, fall in under? It would be valuable to provide a Table that delineates the four classes of mutations suggested, describing the molecular phenotype, physiological phenotype, examples of mutations with references, which connexins are found in each class? Highlight that all mutations cause hearing loss (to what degree?). Under “class 2”, hearing loss is not mentioned. What is the compelling defining difference between class 3 and class 4? Mucositis?
Line 184: Check reference 75, should it be reference 86? Check all references throughout.
Line 190: Rewrite. Move “until the wound is closed”. It can be misunderstood as if the “wound is closed in the granular layer”
Line 207: perhaps “late” 1990s?
Line 210-211: numerate reference “Djallilian et al. Also, is reference 86 correct?
Line 219: It would be valuable to give some insight, or list or describe all epithelia Cx26 is known to be expressed in. For example in a Table. This could also be derived from databases. (bladder, kidney, pancreas, esophagus, cervix, breast, etc. etc.).
Line 220 onwards: The sections go into some detail and speculation of a variety of issues here, but the key messages are not very clear. For example, authors should clarify the idea for the possible role of Cx26 in thicker skin as a mechanism protecting against infections (“environmental damage” is a bit unspecific). This would, however, does not fit in a paragraph named “Intestinal epithelia (remove L) and the auditory system”. Rewrite the paragraph with the general idea of heterozygous advantage and clarify the ideas.
Line 261: provide a reference.
Line 277 onwards: Expand and describe better what is known regarding the role of Cx26 in the cornea and how mutations potentially cause keratitis. What class 1-4 mutations cause keratitis and how?
Line 297: define purinergic signaling for the reader.
Line 306: What is meant by “aggression”
Section 7:
Only 1 sentence mentions Cx26. It would be better to refer to actual clinical trials (clinical trials.gov) rather than mentioning specific companies. “Future directions” could be enhanced. What are the challenges? What is needed to dissect the role of Cx26 in the “tissue barrier”? What is missing in terms of disease mechanism and potential therapeutic opportunities. Is new technology such as CRISPR providing insight?
Figure 1: Consider renaming “AT” to the standard “N-terminus” terminology as used for C-terminus and in the main text. State in Figure legend what the different colored mutations are referring to? (consider having 1 color for each of the four classes of mutations?).
Figure 2: Describe in legend what the connexins in “light grey” signify.
Check all references. There is a mixed use of CX and connexin in the manuscript (connexin is a simple short word so perhaps no need to abbreviate)
Author Response
Reviewer 4
We would like to thank the reviewer for their helpful comments and have summarised our response to comments below in italics for clarity.
Garcia-Vega et al. review the role of connexins and the epithelial barrier with a focus on Cx26. It is not made clear why the authors have chosen to focus on Cx26, especially since the review is overall very general and only very briefly touches upon several broad topics that concern many connexin isoforms. Consequently, the review serves mostly as a very general introduction to the topic as it does not provide too much deep insight into a specific topic or problem or provide a strong point of view or gives a unique insight. Indeed, the general nature of the article is perhaps underscored by the fact that 19 of the first 20 references are reviews themselves.
Some minor suggestions and some moderate issues should be addressed to improve the purpose as a general review that serves to introduce someone to the topic.
In general, the authors focus on Cx26 and “barrier” function. Barrier function needs to be explained in more detail (including the molecular mechanism regulating this phenomenon) as well as the direct evidence that Cx26 truly contributes to barrier function, or not.
We thank the reviewer for their comments. We appreciate that a range of connexins are discussed, predominantly Cx43 and Cx26. However, we have focussed on Cx26 as it is often underlooked and clearly contributes to rapid modulation in a range of tissue. We hope that the changes made throughout the manuscript highlight this. We have included a little extra information on barrier formation including reference to the interplay of tight junctions and connexins, in maintaining epithelial integrity.
Specifics:
Line 13: rephrase:
The wording “external environment” may be misleading to the reader (it may not be obvious this only refers to external to the organ itself). Consider rephrasing keeping in mind that epithelial cells cover the inner surface of the internal organs and the outer surface of the human body.
We have addressed this by referring to tissue lining the lumen of organs connecting with the outside environment.
Line 17: For clarity consider deleting or moving “Within the epithelia” (no “L” at the end) to after “protein function”
We have modified this sentence
Line 31: rewrite. Purinergic signaling is not a “physiological process”. (Simplest rewrite would be “…wound healing VIA purinergic signaling…”)
We have removed purinergic signalling from this sentence
Line 32: Correct the erroneous statement that Cx26 is the smallest member of the family.
This has been modified
Line 33: Add that Cx26 is also associated with keratitis (eye disease of another type of epithelia).
This has been added and referred to in the section on the cornea
Line 44: rephrase confusing sentence considering epithelial (endothelial) cells are lining what is considered the “inner” surface of blood vessels.
Reference to endothelial cells and blood vessels has been removed
Line 49: rephrase to avoid confusion. Although gap junctions would play an important role in avascular epithelia, other pathways are providing for example oxygen and glucose delivery, it is not a gap junction-exclusive exchange of signals.
We appreciate this concept and have rephrased this sentence
Line 62-78: Section 2 is a short and very general paragraph that perhaps could be included in the general introduction that describes the epithelium. Or perhaps expand the section. In any case, highlight that there are many additional important functions of the epithelium apart from barrier function (e.g. glandular secretion).
We have divided the introduction section into two short units – thereby reducing the number of subsections. We have also included information on glandular secretion and hair follicles etc, although a role for connexin signalling in these events is not discussed in the present review
Line 83: Specify to the reader that keratinocytes are epithelial.
This has been clarified
Line 109 onwards: In the section of connexins in the skin, the expression of Cx26 is not very well detailed. Is Cx26 truly expressed in normal non-irritated skin? or only palmoplantar skin? What about Cx26 in the keratinocytes in hair follicles and sebaceous gland epithelial cells?
Cx26 is expressed and documented in normal human skin, all be it at low levels. CX26 is expressed in keratinocytes and readily detected by immunofluorescence in skin biopsies and in 3D organotypic models. We have added a few further references here.
The section that follows on translational regulation and trafficking (and associated Figure 3) is not particularly relevant to the topic or the section “connexins in the skin”. It would valuable to add some information on the actual transcriptional regulation (eg AP1 and other transcription factors) that underlies the dramatic upregulation of Cx26 under certain situations. This can be linked to certain inducing pathways such as interleukins etc. as discussed later.
We appreciate the comment here and have included a short section on transcriptional control of Connexin 26. API and NfKB binding sites are key examples.
Line 110: Confirm whether Cx26 has the shortest C-tail (shorter than Cx23 and Cx25?)
We have revised this statement and appreciate the concepts of the more recently identified but less studied Cx23 and Cx25
Line 124-6: the phrases “standard” and “resting” conditions are not ideal.
These phrases have been reworded
Line 145 onwards. Are the four classes of mutations a classification suggested by the authors? Specify and if not provide a reference. What is determining these mutation classes? The molecular effect (eg trafficking) or the phenotype? Or both? Which class does for example R143W, which causes reduced cell death, fall in under? It would be valuable to provide a Table that delineates the four classes of mutations suggested, describing the molecular phenotype, physiological phenotype, examples of mutations with references, which connexins are found in each class? Highlight that all mutations cause hearing loss (to what degree?). Under “class 2”, hearing loss is not mentioned. What is the compelling defining difference between class 3 and class 4? Mucositis?
We have slightly reworded this section. The class 4 mutation were proposed by us in two recent manuscripts. R143W is a recessive mutation associated with hearing loss and is a class one mutation with no trafficking deficiency but loss of function this is now referred to in the relevant section..
Line 184: Check reference 75, should it be reference 86? Check all references throughout.
We have rechecked the references, thank you for pointing this out
Line 190: Rewrite. Move “until the wound is closed”. It can be misunderstood as if the “wound is closed in the granular layer”
We have revised this sentence
Line 207: perhaps “late” 1990s?
Thank you yes we agree to this change
Line 210-211: numerate reference “Djallilian et al. Also, is reference 86 correct?
We have amended this
Line 219: It would be valuable to give some insight, or list or describe all epithelia Cx26 is known to be expressed in. For example in a Table. This could also be derived from databases. (bladder, kidney, pancreas, esophagus, cervix, breast, etc. etc.).
We have included a comprehensive table enabling us to highlight the rationale for focus on connexin 26
Line 220 onwards: The sections go into some detail and speculation of a variety of issues here, but the key messages are not very clear. For example, authors should clarify the idea for the possible role of Cx26 in thicker skin as a mechanism protecting against infections (“environmental damage” is a bit unspecific). This would, however, does not fit in a paragraph named “Intestinal epithelia (remove L) and the auditory system”. Rewrite the paragraph with the general idea of heterozygous advantage and clarify the ideas.
We have revised this paragraph and trust that it highlights the concept of the relationship between the link of connexin26 in auditory function and the role in intestine with putatitve links to heterozygous advantage
Line 261: provide a reference.
This is now provided
Line 277 onwards: Expand and describe better what is known regarding the role of Cx26 in the cornea and how mutations potentially cause keratitis. What class 1-4 mutations cause keratitis and how?
A link here has been provided
Line 297: define purinergic signaling for the reader.
This is defined
Line 306: What is meant by “aggression”
This has been reworded
Section 7:
Only 1 sentence mentions Cx26. It would be better to refer to actual clinical trials (clinical trials.gov) rather than mentioning specific companies. “Future directions” could be enhanced. What are the challenges? What is needed to dissect the role of Cx26 in the “tissue barrier”? What is missing in terms of disease mechanism and potential therapeutic opportunities. Is new technology such as CRISPR providing insight?
We have made some further comments here and added some additional examples of the potential of CX26 therapies
Figure 1: Consider renaming “AT” to the standard “N-terminus” terminology as used for C-terminus and in the main text. State in Figure legend what the different colored mutations are referring to? (consider having 1 color for each of the four classes of mutations?).
This has been amended
Figure 2: Describe in legend what the connexins in “light grey” signify.
This has been amended so as all are in black
Check all references. There is a mixed use of CX and connexin in the manuscript (connexin is a simple short word so perhaps no need to abbreviate)
We have double checked all references and only used CX for human connexin anywhere else we have used connexin